# Acrylic Bone Cement Incorporated with Low Chitosan Loadings

**DOI:** 10.3390/polym12071617

**Published:** 2020-07-21

**Authors:** Mayra Eliana Valencia Zapata, José Herminsul Mina Hernandez, Carlos David Grande Tovar

**Affiliations:** 1Grupo de Materiales Compuestos, Escuela de Ingeniería de Materiales, Universidad del Valle, Calle 13 #, Cali 100-00, Colombia; valencia.mayra@correounivalle.edu.co; 2Grupo de Investigación de Fotoquímica y Fotobiología, Universidad del Atlántico, Carrera 30 Número 8-49 Puerto Colombia 081008, Colombia

**Keywords:** acrylic bone cement, bioactivity, biocompatibility, chitosan, poly (methyl methacrylate)

## Abstract

Despite the potential of acrylic bone cement (ABC) loaded with chitosan (CS) for orthopedic applications, there are only a few in vitro studies of this composite with CS loading ≤ 15 wt.% evaluated in bioactivity tests in simulated body fluid (SBF) for duration > 30 days. The purpose of the present work was to address this shortcoming of the literature. In addition to bioactivity, a wide range of cement properties were determined for composites with CS loading ranging from 0 to 20 wt.%. These properties included maximum exotherm temperature (*T_max_*), setting time (*t_set_*), water contact angle, residual monomer content, flexural strength, bending modulus, glass transition temperature, and water uptake. For cement with CS loading ≥ 15 wt.%, there was an increase in bioactivity, increase in biocompatibility, decrease in *T_max_*, increase in *t_set_*, all of which are desirable trends, but increase in residual monomer content and decrease in each of the mechanical properties, with each of these trends, were undesirable. Thus, a composite with CS loading of 15 wt.% should be further characterized to explore its suitability for use in low-weight-bearing applications, such as bone void filler and balloon kyphoplasty.

## 1. Introduction

For years, acrylic bone cement (ABC) has played a pivotal role in orthopedic surgery. ABC is used to anchor total joint replacements (TJRs), remodel areas, and in the stabilization and augmentation of osteoporosis-induced vertebral body fractures (vertebroplasty and balloon kyphoplasty) [1]. ABC is biocompatible, and, in a TJR, it provides fast fixation between the bone and the implant. However, ABC has some disadvantages, such as a high maximum exothermic temperature (*T_max_*) [2], a lack of bioactivity [3,4,5], and monomer toxicity [6].

Bioactive materials are those that can stimulate a biological response in their environment. These are classified into three groups: Osteoconductive are materials that stimulate bone growth on the surface; osseointegration are materials that can stimulate the growth of new bone, forming a stable anchorage with the bone, and osteoinductive substances can induce bone formation in parts where the bone does not naturally grow [7,8].

Natural polymers attract attention for use in biomaterials because they possess significant similarity to biological structures, so they are readily accepted by the tissue environment and, therefore, quickly metabolized, obtaining non-toxic by-products easy to eliminate [9].

Chitosan (CS) is a copolymer of glucosamine and N-acetylglucosamine deacetylated derived from chitin [9], the second most abundant natural polysaccharide on earth (found in the shells of crabs, shrimps, and insects, for example) [7]. CS-based materials are ideal bioactive materials due to their biocompatibility, wound-healing ability, non-toxicity, antibacterial properties, anti-fungal properties, hydrophilicity, biodegradability, and ability to stimulate adherence, proliferation, and viability of cells [7,9,10]. Several reports exist on the addition of CS to ABC, resulting in improvement of various properties of ABC, such as increased setting time (*t_set_*) [11], reduced maximum exotherm temperature (*T_max_*) [6], increased porosity, higher antibacterial action [12], and increased bioactivity [2,5,13,14,15]. However, high CS loading generates considerable detriment to the mechanical properties of cements, restricting clinical uses to non-load-bearing applications. It is crucial to study lower concentrations of CS to establish an adequate balance between biological and mechanical properties that will allow it to expand its clinical usefulness to a wide range of low-weight-bearing applications such as kyphoplasty, cranioplasty, bone defect fillers, maxillofacial reconstruction, and others [16,17]. 

The reported bioactivity of cements CS-loaded in a simulated biological fluid (SBF) or buffered phosphate solution (PBS) was only obtained during short test times (<30 days) with cement formulations that have a high CS loading (mass of CS as a proportion of the weight of the solid component of the cement) (>20 wt.%). Information on hydrolytic degradation, bioactivity, and other properties of ABC with CS content ≤ 20 wt.%, under simulated biological fluid conditions after prolonged test times (>30 days), is lacking. 

The purpose of the present study was to determine the influence of CS loading on hydrolytic degradation, bioactivity, and several handling, physical, and mechanical properties of ABC loaded with ≤ 20 wt.% CS, with testing in SBF over 28 weeks.

## 2. Materials and Methods

### 2.1. Materials

Methyl methacrylate (MMA), 2-(diethylamino) ethyl acrylate (DEAEA), 2-(diethylamino) ethyl methacrylate (DEAEM), benzoyl peroxide (BPO), and chitosan from shrimp shells with a molecular weight between 190–310 KDa and deacetylation degree of 88% were all purchased from Sigma-Aldrich (Sigma-Aldrich, Palo Alto, CA, USA). *N,N*-dimethyl *p*-toluidine (DMPT) was purchased from Merck (Merck, Burlington, MA, USA). Barium sulfate was purchased from Alfa Aesar (BaSO_4_) (Alfa Aesar, Tewksbury, MA, USA). PMMA powder beads were purchased from Veracryl (New Stetic SA, Medellin, Colombia).

### 2.2. Preparation of ABC

ABCs with different CS loadings were prepared. The solid-phase (SP) and liquid phase (LP) were made separately by hand mixing, according to the composition shown in Table 1. LP in all formulations was comprised of 95.5 wt.% MMA, 2.0 wt.% of a 50:50 mixture of the comonomers (DEAEA:DEAEM), and 2.5 wt.% DMPT as the activator of the polymerization reaction. LP was added in SP at a solid/liquid ratio = 2 g/mL and manually mixed. The resulting dough was transferred to Teflon molds to obtain specimens to be used in the different tests.

### 2.3. Characterization Physical, Chemical and Thermal of ABC

ABCs specimens were characterized by Fourier transform infrared (FTIR) in attenuated total internal reflectance mode (ATR) in a Spectrum One spectrometer (Perkin-Elmer, Waltham, MA, USA). Thermogravimetric analysis (TGA) was carried out in a TGA Q500 analyzer (TA Instruments, New Castle, DE, USA) in a range of 50–500 °C under nitrogen at a heating rate of 10 °C/min. Differential scanning calorimetry (DSC) was performed in a DSC 8500 (Perkin-Elmer, Waltham, MA, USA) in a range of −20–150 °C under nitrogen at a heating rate of 10 °C/min.

The water contact angle (WCA) was measured at 25 °C using a KSV CAM200 tensiometer (KSV Instruments, Helsinki, Finland). At least 10 separate measurements were taken on each sample.

Proton nuclear magnetic resonance spectra (^1^H–NMR) were carried out in a Bruker Avance III HD-400 spectrometer (Bruker, Billerica, MA, USA) at 25 °C. Specimens were dissolved in deuterated chloroform (CDCl_3_) one week after being prepared. Residual monomer content (RMC) in a sample was calculated by integrating the signals of the methoxy protons of the MMA and the PMMA using Equation (1).
(1)$$\mathrm{RMC}\left(\%\right)=\frac{A_{MMA}}{A_{MMA}+A_{PMMA}}\times 100$$
where *A_MMA_* is the signal area of methoxy protons of *MMA* (δ = 3.7 ppm), and *A_PMMA_* is the signal area of methoxy protons of PMMA (δ = 3.5 ppm).

### 2.4. Determination of Handling Properties 

Handling properties of the cement paste were evaluated following the protocols given in ISO 5833-02 [18]. After the mixing of the ABC components, the cement was deposited in a Teflon mold (diameter and height of 68 and 20 mm, respectively) to record time and temperature data. Temperature against time plot allowed us to calculate the maximum temperature. Setting time (*t_set_*) was calculated like time required to reach setting temperature (*T_set_*) calculated according to equation 2. The test was carried out in duplicate.
(2)$$T_{set}=\frac{T_{max}+T_{amb}}{2}$$
where *T_amb_* is the recorded ambient temperature, and *T_max_* is the highest temperature attained.

### 2.5. Mechanical Properties

Four-point bending and compression tests were conducted using a universal testing machine Tinius Olsen-H50KS (Tinius Olsen, Redhill, United Kingdom), following the protocols given in ISO 5833-02 [18]. For the former test, specimens measuring 75 mm × 10 mm × 3 mm were tested at a crosshead displacement rate of 5 mm/min. The bending strength (B) and modulus (E) were calculated using Equations (3) and (4), respectively.
(3)$$B=\frac{3Fa}{bh^{2}}$$
(4)$$E=\frac{\Delta Fa}{4fbh^{3}}x\left(3l^{2}-4a^{2}\right)$$

For Equation (3), *F* is the force at break, b is the average measured width of the specimen, h is the average measured thickness of the sample, and a is the distance between the inner and outer loading points. For Equation (4), Δ*F* is the load range (50 N − 15 N = 35 N), *f* is the difference the deflections under loads of 15 and 50 N, and l is the distance between the outer loading points.

For the compression test, cylindrical specimens (diameter and height of 6 and 12 mm, respectively) were tested at a crosshead displacement rate of 20 mm/min. For each of these tests, a minimum of six specimens was tested for each of the cement formulations.

The morphology of the surfaces of compression test specimens, at the end of the test, were observed using an environmental scanning electron microscope (ESEM) Philips XL30 (Philips, Eindhoven, The Netherlands).

The dynamic mechanical analysis was carried out in a DMA Q800 (TA Instruments, New Castle, DE, USA) in Dual Cantilever mode to 1 Hz of frequency in a range from −100 to 200 °C at a heating rate of 3 °C/min. The test samples were prepared with the dimensions of 46 × 6 × 4 mm^3^. At least three specimens were tested per formulation. 

### 2.6. Assessment in a Simulated Biological Fluid (SBF)

Hydrolytic degradation of cement specimen and bioactivity (formation of an apatite-like layer on cement specimen) were determined using disc specimens (diameter = 6 ± 0.1 mm and a thickness = 2 ± 0.1 mm) and an SBF that was prepared using the methodology presented by Kokubo and Takadama [19], with its ionic concentration being similar to that of human blood plasma. On the other hand, samples to monitor compressive strength had a thickness of 12 ± 0.1 mm. 

#### 2.6.1. Hydrolytic Degradation

The protocols used followed those given in ASTM F1635-16 [20]. Cylinders were immersed in 50 mL of SFB to 37 °C in an incubator (Memmert, Schwabach, Germany) for periods between 1 and 28 weeks. At each period, at least three cylinders were evaluated. The solution was replaced with fresh SBF weekly. During the time of evaluation, the pH of the SBF was recorded using an Accumet ™ AB150 pH meter (Fisherbrand, Ottawa, ON, Canada). Before immersion in the test solution, the specimen was weighed (*W*_0_). At the end of the test period, the samples were removed from the SBF, washed with distilled water, dried superficially with a dry tissue, and weighed (*W_w_*). They were then dried, in air, at 37 °C, for 72 h and weighed again (*W_d_*). Hence, the weight loss (*W_l_*) and water uptake (*W_u_*) of the specimen were determined by Equations (5) and (6), respectively. Compressive strength was evaluated under the same conditions described in the mechanical tests.
(5)$$W_{l}\;\left(\%\right)=\frac{W_{0}-W_{d}}{W_{0}}\times 100$$
(6)$$W_{u}\;\left(\%\right)=\frac{W_{w}-W_{d}}{W_{d}}\times 100$$

#### 2.6.2. Bioactivity in SBF

The morphology of the surface of a specimen used in the hydrolytic degradation test was obtained using a scanning electron microscope (SEM). The attached X-ray energy dispersive spectrometer (EDS) JEOL Model JSM 6490 LV (Akishima, Tokyo, Japan) supported calculation of the amount of apatite-like deposition layer (that is, proportional to the amount of Ca and P ions on the surface).

### 2.7. Statistical Analysis

Quantitative property results are presented as mean (standard deviation). For each of these properties, the test of significance of the difference between means was performed using the Student t-test with the software Minitab 17 (Minitab, LLC, State College, PA, USA). The difference was considered statistically significant when *p* < 0.05. Statistical analysis was performed comparing ABCs loaded with respect to control ABC.

## 3. Results

Currently, PMMA-based bone cement is the most widely used type of bone cement in orthopedics [6]. In this research, ABC loaded with CS between 0–20 wt.% in its solid phase were prepared. The influence of CS content on mechanical, thermal, chemical, physical, and biological properties were analyzed.

### 3.1. FTIR

Figure 1 shows the FTIR spectra of the CS-loaded ABCs. The bands at 1140 and 1720 cm^−1^ corresponds to O–CH_3_ stretching vibration and C=O asymmetric stretching vibrations, respectively, corresponding to the PMMA structure [21]. The cement containing CS presented a wide overlapped band corresponding to the amine N–H stretching vibration (3365 cm^−1^) and the O–H stretching vibration (3460 cm^−1^) in the CS. Characteristics bands of PMMA and CS without shifting indicate that the preparation conditions of ABC did not promote a covalent bonding between the chitosan and the cement. Therefore, the interaction between them was only governed by physical forces. 

### 3.2. TGA

The thermal stability of cement is improved with the incorporation of CS (Figure 2). The temperature at which a 10% loss of weight (*T*_10_) is achieved is reduced from 304 °C for ABCs without CS, to 288 °C for ABCs loaded with 20 wt.%. Additionally, *T*_50_ changed from 377 to 380 °C for ABCs without CS and ABC loaded with 20 wt.%, respectively. A 20 wt.% of CS introduction decreased the total loss weight percentage to 4 °C. At temperatures > 400 °C, a significant weight loss of the formulations with lower CS contents is evident. Figure 2 also shows an inset (a derivative of the thermogravimetric analysis, DTGA), in which the peak of the curve represents the temperature of the highest rate of weight loss obtained.

### 3.3. Handling Properties and Water Contact Angle

The handling parameters that describe the polymerization reaction are maximum temperature (*T_max_*) and setting time (*t_set_*). *T_max_* corresponds to the maximum temperature reached during the MMA free radical polymerization reaction, which is exothermic, and the *t_set_* represents the time taken to reach a temperature midway between ambient and *T_max_*. From *t_set_*, the significant increase of paste’s viscosity caused a reduction in the handling of the cement [22]. 

The maximum temperature obtained during the MMA polymerization reaction shows a decrease of 13 °C with a CS loading of 20 wt.% (Table 2). The increase in *t_set_* is significant for formulations with CS loading ≥ 15 wt.%. For each of the formulations, *T_max_* and *t_set_* values are within limits allowed by ISO 5833-02 for ABC [18]. 

The water contact angle showed a decrease of 7.1% and 12.5% with CS loadings of 15 and 20 wt.%, respectively (*p* < 0.01). Lower WCA is related to a more significant wetting on the ABC surface due to an increase in hydrophilicity. Other researchers have found WCA values similar to those found in this study [23].

### 3.4. Mechanical Properties, Residual Monomer Content and DSC

The bending strength, bending modulus, and compressive strength results are presented in Figure 3. ISO 5833-02 [18] specifies a minimum value of bending strength, bending modulus, and compressive strength for ABC as 50, 1800, and 70 MPa, respectively. Only the formulation without CS achieved the minimum bending strength. Except for the composite with CS of 20 wt.%, all other formulations produced the minimum bending modulus. Formulations with CS loadings of 15 and 20 wt.% did not achieve the minimum compressive strength. 

Both porosity and surface roughness of the compression test specimens increased with an increase in CS loading (Figure 4), indicating high potential for cell attachment for these specimens.

Storage modulus (*E*’) obtained by DMA represents the elastic phase of the complex modulus and is equivalent to the energy stored through deformation [24]. It is related to the elastic modulus in bending; therefore, according to Figure 5, the stiffness of the cement decreased with increasing CS loading. *T_g_* was calculated from the maxima value of tan δ versus plotted temperature, and the trends in *T_g_* determined by DSC and DMA were similar (Table 3).

From Table 3, it can be observed that *T_g_* determined by DSC and DMA show similar behavior, in which CS loading ≤ 10 wt.% increases the *T_g_*, while rates above this value generate a slight decrease in this property. There is a small increase in RMC with an increase in CS loading, except for the formulation with 20 wt.% CS, which offers an increase of more than 400% (Table 3). High RMC suggests a high risk of chemical necrosis. 

### 3.5. Hydrolytic Degradation

Figure 6 shows the change in weight, water uptake, pH, and compressive strength of ABCs with immersion time duration in SFB. The results showed that weight loss (Figure 6a) and water uptake (Figure 6b) increased with an increase in CS loading. The absorption is favored by the increase in the porosity of the cement, increasing the weight loss generated possibly by the hydrolytic degradation of CS within the cement.

The pH of the SBF (Figure 6c) showed a decrease over time, possibly due to the CS degradation products, while when plain cement (no CS) test specimens were used, the change in SBF pH during the test was marginal. For each of the formulations, the variation of compressive strength with immersion time in SBF was negligible (Figure 6d), due to the few weight loss reported for these cement specimens (<2.5%) (Figure 6a).

### 3.6. Bioactivity in SBF

The presence of Ca and P on the surface of the ABCs immersed in SFB was evaluated by SEM and EDS techniques. Bioactivity in SBF was determined by monitoring the deposition of these ions on the surface of the ABC.

#### 3.6.1. SEM

The morphologies of the surfaces of the cement specimens are shown in Figure 7. It is seen that the surface roughness increases with an increase in CS loading. For the same formulation, there are no apparent differences in surface morphology with an increase in immersion time in SBF. 

#### 3.6.2. EDS

Formulations with CS loadings of 15 and 20 wt.% CS show Ca deposition from the first week of immersion and an increase in the content of this ion until week 8, by which time the entire surface of the cement is covered (Figure 8a). On the other hand, for the same formulations, P appears from week 4 (Figure 8b). In contrast, for formulations with a CS loading < 15 wt.%, the deposition of these ions is delayed for several weeks (Figure 8a,b). 

It is known that bioactivity in SBF is related to the facility of deposition of Ca and P ions on the surface of the cement, which, under in vivo conditions, promotes cell attachment of osteoblasts during the process of bone remodeling. The present results show that bioactivity in SBF improved with an increase in CS loading. The contact angle results corroborate these findings (increase in hydrophilicity with a rise in CS loading) (Table 2) and the surface morphologies (increase in the amount of surface CS with an increase of CS loading) (Figure 7). 

## 4. Discussion

The results of the chemical, physical, thermal, mechanical, and biological characterizations of the ABCs prepared in this investigation were analyzed according to the effect of CS content. 

The lack of chemical interaction between CS and PMMA, as confirmed by FTIR, shows that CS is not modified and, as such, it is expected that CS will impart some of its attractive properties, such as biocompatibility, bioactivity, and biodegradation [9,25], to the cement. 

From the mechanical point of view, this lack of chemical bonding drastically affected the bending and compressive properties of the cements since the interface formed between them is weak. On the other hand, the thermal stability of the ABCs determined by TGA was improved at high temperatures, while at low temperatures, it had no significant influence.

The rigid glucosamine structure and the semi-crystalline structure of CS caused an increase in the cement viscosity [13,15]. This increased viscosity means that during the polymerization reaction, voids that are formed as entrapped air unable to escape, and the porosity of CS-loaded cement increased [6]. The increased porosity of ABCs with CS loading observed by SEM generates a loss in the mechanical properties. Still, it improves the bioactive behavior because a roughness increased with the biocompatibility favoring cell adhesion and bone tissue growth through the ABC [6,23]. It has been reported that the bone ingrowth to the ABC structure forms a strong mechanical interlocking between the cement and bone, and thus provides stability to the implant site [26]. 

The decrease in the maximum temperature improves the performance of the ABCs because it decreases the risk of thermal necrosis of the tissue around the cement. The increase in the setting time gives the surgeon more time available for filling and remodeling bone defects.

The trend of decrease in Tmax with an increase in CS loading, which is the same as reported in the literature [13,15,23], is attributed to the intramolecular hydrogen bonds in CS acting as a heat sink, that is, absorbing the released heat during polymerization of the cement [3,13,15]. The reduction in temperature during the polymerization reduces the risk of bone cell necrosis and subsequent implant loosening [3]. Furthermore, the increase in *t_set_* is related to the longer time available for the surgeon to manipulate the ABC during surgery.

The biocompatibility of a material is related to the surface properties, such as hydrophilicity [23]. In this research, the hydrophilicity of ABCs surfaces was determined by the water contact angle. For cement with CS loading ≤ 10 wt.%, WCA increased with an increase in CS loading, which is attributed to the surface of these cements being covered with a layer of the PMMA produced during the polymerization of MMA. In contrast, with CS loading > 10 wt.%, WCA decreases with an increase in CS loading because of the more significant number of hydrophilic groups (NH_2_ and OH) provided by the CS. The decrease of the contact angle favored the biocompatibility in SBF in ABC with 15 and 20 wt.% [3]. 

The significant decrease in each of the mechanical properties with an increase in CS loading may be attributed to two factors, namely (1) lack of chemical bonding, which means that the interface between the cement matrix and CS is weak; and (2) increased porosity. The low mechanical performance of CS-loaded cements concerning ISO 5833-02 [18] indicates that they should not be used for prosthesis fixation [27], but rather in applications such as remodeling areas of metastatic or osteoporotic cancer, in fractures, in the repair of cranial defects (cranioplasties) or in the restoration of vertebral fractures (vertebroplasty and kyphoplasty) [1], where less mechanical performance is required.

The *T_g_* affects the mechanical properties of ABCs. It has been reported that the *T_g_* of bone cements shows a direct relationship with modulus [24,28]. Our results confirm that the highest *T_g_* was obtained for formulations with percentages ≤ 10 wt.%, which is related to the mechanical properties. 

In general, the residual monomer content of formulations with percentages of CS less than 20 wt.% was low; however, the formulation with CS loading of 20 wt.% presented a high value compared to the other formulations. This increase in the RMC with the CS loading may be due to the lower diffusion capacity of the monomer during polymerization, generating more excellent heat dissipation [29]. Kühn Dieter [30] reports RMC between 1.8–4.1% for some of the available commercial formulations, so it can be assumed that the formulation with CS loading of 20 wt.% could present a risk of toxicity of the residual monomer [31]. 

The high weight loss and high-water uptake of ABCs with CS loading > 15 wt.% after soaking in SFB could be attributed to CS being hygroscopic and, as such, promotes the absorption of water. Also, higher porosity of the material, due to chitosan degradation, could encourage uptake water of the cement [6]. Similar weight loss and water uptake results have been reported elsewhere [6]. The pH of the SFB decreased over time, indicating that the by-products of the degradation process are responsible for the pH of the fluid [32,33]. For a given cement formulation, the change in its compressive strength over time was negligible, a trend that is consistent with the weight loss profile. In turn, this profile may be attributed to the reduced solubility of CS at pH > 6.5 [9,23].

Bioactivity relates to a material’s ability to stimulate a biological response in its environment [7]. In this research, bioactivity in SBF was determined by Ca and P ions deposition on the cement surface. These ions are the main inorganic components of the bone, which, once deposited on the extracellular matrix material produced by the osteoblasts, generate the osteoid mineralization and, therefore, the formation of new bone [34].

The presence of Ca ions was confirmed one week later of immersion in SFB with CS loadings ≥ 15 wt.%. On the other hand, P presence was confirmed only after the fourth week. The previous results coincide with the sequence of processes reported for the formation of apatite. First, Ca^2+^ ions in SFB are selectively adsorbed onto negatively charged surfaces (OH^−^ groups of the CS). Second, PO_4_^3−^ ions are attracted to the calcium-rich surfaces to form an amorphous calcium phosphate (ACP), and finally, the ACP is converted into apatite [9,13,17,35]. The presence of Ca and P ions in the first week of immersion of the cement specimens in SFB has been reported in the literature for cement formulations with CS loading > 20 wt.% [3,13,23]. 

Formulations with chitosan loading < 15 wt.% also showed bioactivity in SBF. ABCs CS loading of 10 wt.% started the deposition process after second-week soaking, while those with 0 and 5 wt.% CS registered bioactivity from week 12. For cements without CS loading, bioactivity in SBF may be attributed to the presence of alkaline comonomers [36].

The results obtained were promising for this type of cement with CS loading ≤ 20 wt.%; therefore, a more extensive biological characterization including in vitro and in vivo conditions is proposed in future research.

## 5. Conclusions

Formulations with CS loading ≥ 15 wt.% showed deposition of Ca from the first week following immersion in SBF and deposition of P ions from the fourth week, in contrast to cements with lower CS loading, which presented deposition between the second and twelfth week. This observation might suggest that for fast bioactivity in SBF, the minimum CS loading is 15 wt.%.

Formulation with CS loading ≥ 15 wt.% showed increased thermal stability, higher porosity, increased roughness, decreased contact angle (on water), reduced *T_max_*, and increased tset. Those results were desirable, but this was accompanied by increased RMC, and significant decreases in bending strength, bending modulus, compressive strength, and storage modulus, each of which is undesirable. Thus, the cements with CS loading of 15 wt.% should be further characterized for use in low-weight-bearing applications, such as bone void fillers, vertebroplasty, and balloon kyphoplasty. 

## Figures and Tables

**Figure 1 polymers-12-01617-f001:**
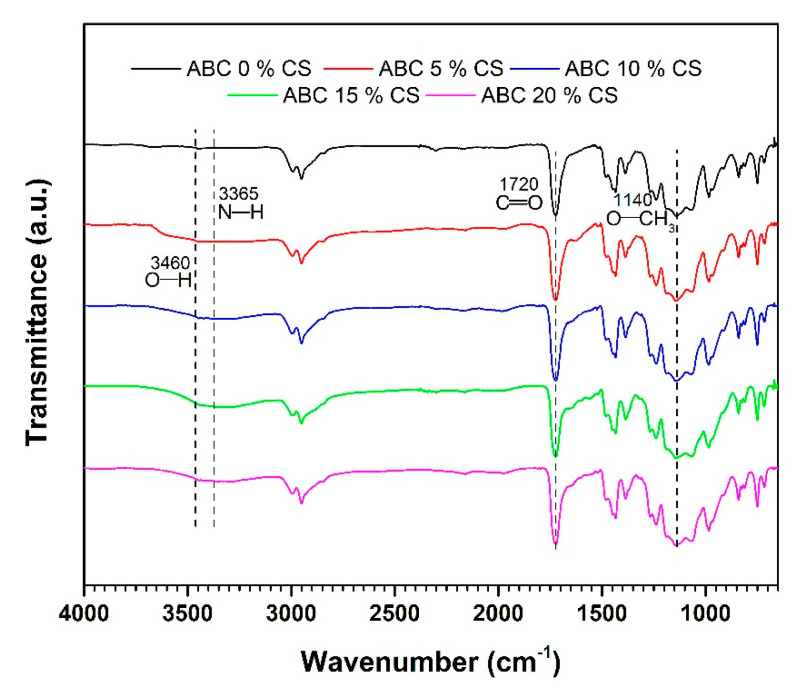
Fourier-transform infrared spectra in attenuated total internal reflectance mode (ATR) mode for ABCs loaded with chitosan (CS).

**Figure 2 polymers-12-01617-f002:**
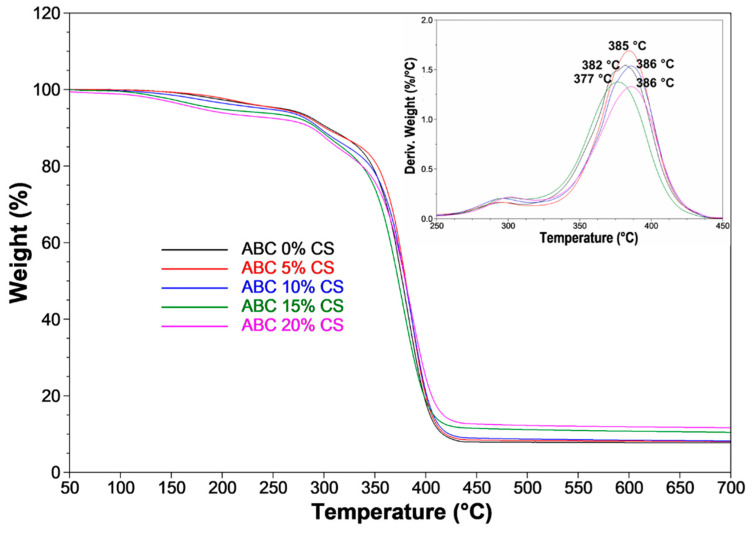
Thermogravimetric analysis (TGA) for ABCs loaded with CS. The inset shows the derivative TGA (DTGA) thermograms.

**Figure 3 polymers-12-01617-f003:**
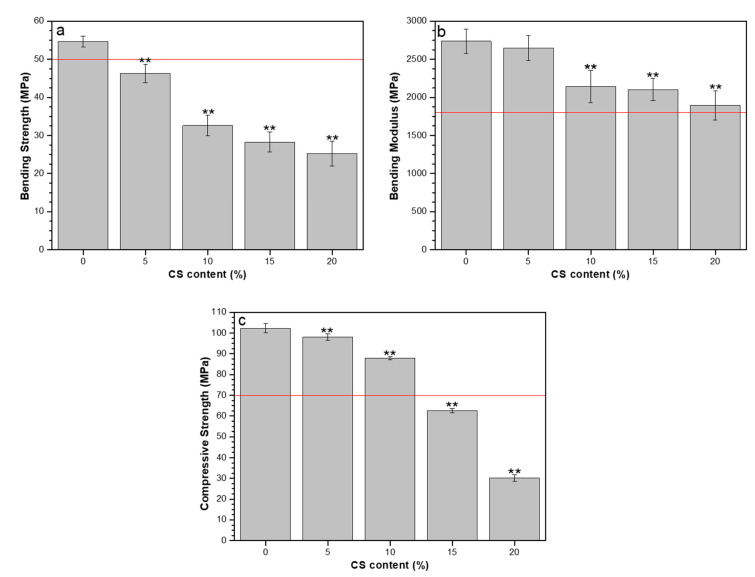
Mechanical properties in ABCs loaded with CS: (**a**) bending strength, (**b**) bending modulus, and (**c**) compressive strength. The straight red line corresponds to the minimum values specified in ISO 5833-02. Significance level of * *p*-value < 0.05; ** *p*-value < 0.01.

**Figure 4 polymers-12-01617-f004:**
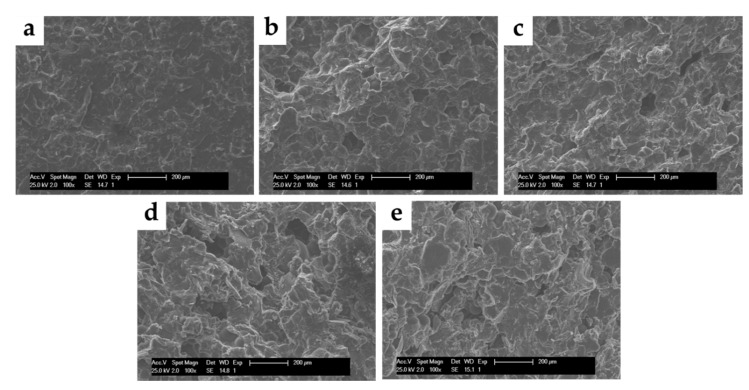
SEM micrographs (100×) of surfaces of ABCs loaded with CS. (**a**) ABC 0% CS, (**b**) ABC 5% CS, (**c**) ABC 10% CS, (**d**) ABC 15% CS, and (**e**) ABC 20% CS.

**Figure 5 polymers-12-01617-f005:**
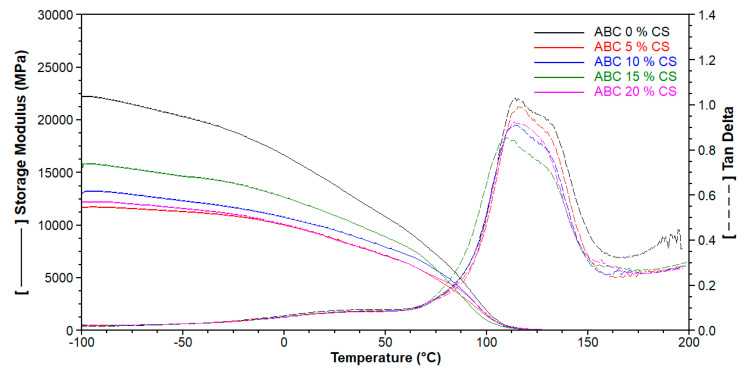
Dynamic mechanical analysis (DMA) of ABCs loaded with CS to 1 Hz of frequency.

**Figure 6 polymers-12-01617-f006:**
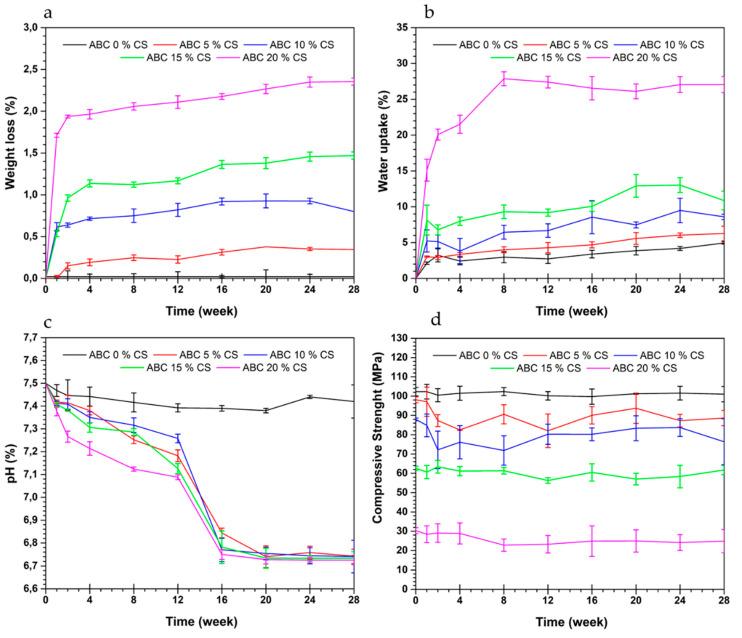
Effect of soaking time in a simulated body fluid of acrylic bone cements loaded with CS on (**a**) weight loss, (**b**) water uptake, (**c**) pH, and (**d**) compressive strength.

**Figure 7 polymers-12-01617-f007:**
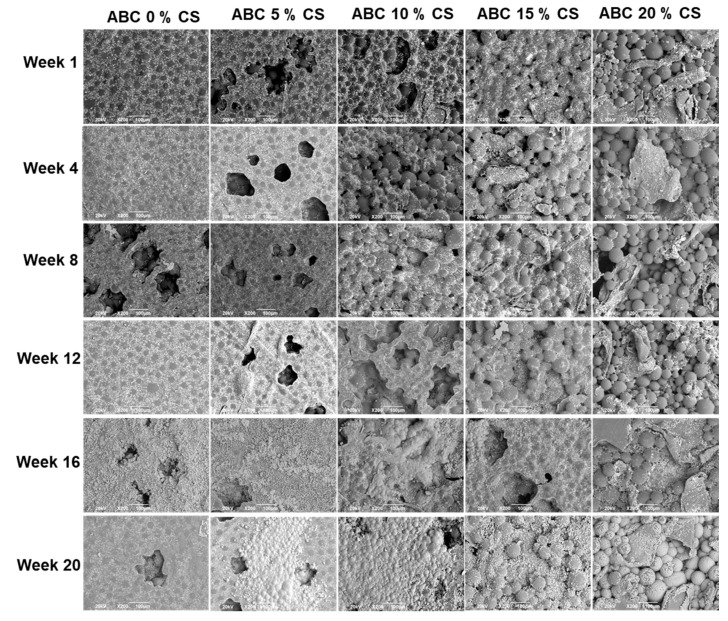
SEM micrographs of the surfaces of ABCs loaded with CS after different times of soaking in simulated body fluid (SFB) (200×).

**Figure 8 polymers-12-01617-f008:**
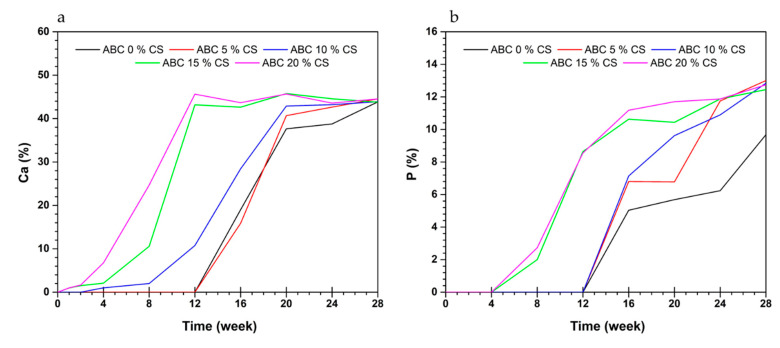
Content of (**a**) Ca and (**b**) P by EDS analysis of the surfaces of ABCs loaded with CS after different times of soaking in SFB.

**Table 1 polymers-12-01617-t001:** Composition percentage (wt.%) of the solid phase (SP) of the acrylic bone cements (ABCs) formulations.

Formulation	SP			
	PMMA	BaSO_4_	BPO	CS
ABC 0% CS	88	10	2	0
ABC 5% CS	83	10	2	5
ABC 10% CS	78	10	2	10
ABC 15% CS	73	10	2	15
ABC 20% CS	68	10	2	20

**Table 2 polymers-12-01617-t002:** The maximum exotherm temperature (*T_max_*), setting time (*t_set_*), and water contact angle (WCA) results. Data reported as mean ± standard deviation.

Formulation	*T_max_* (°C)	*t_set_* (s)	WCA (°)
ABC 0% CS	59 ± 3	355 ± 30	64.9 ± 2.2
ABC 5% CS	58 ± 2	370 ± 15	65.1 ± 1.1
ABC 10% CS	54 ± 3	420 ± 15	66.4 ± 1.4
ABC 15% CS	54 ± 4	460 ± 20 *	60.3 ± 2.1 **
ABC 20% CS	46 ± 2 *	545 ± 25 *	56.8 ± 1.9 **

Significance level of * *p*-value < 0.05; ** *p*-value < 0.01.

**Table 3 polymers-12-01617-t003:** Residual monomer content (RMC) and glass transition temperature (*T_g_*) by DSC and DMA of ABC.

Formulation	RMC (%)	*T_g_*_DSC_ (°C)	*T_g_*_DMA_ (°C)
ABC 0% CS	1.06	106	114
ABC 5% CS	1.83	108	116
ABC 10% CS	1.32	107	115
ABC 15% CS	1.57	105	110
ABC 20% CS	6.25	101	113

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
