# Peer review of "Acrylic Bone Cement Incorporated with Low Chitosan Loadings"

_polymers, 2020, doi:10.3390/polym12071617_

Round 1
Reviewer 1 Report
- Please explain “maximum exotherm temperature” and “setting time” as they can be not well-known terms for Polymers readers
- The preparation procedure is unclear – first (line 77) solid constituents (ref. Table 1) were mixed – it suggests that CS and cement powder while next liquid to solid ratio is given. Additionally, molar/volume/mass ratio?
- 6: in my opinion, if the initial mass of the tested sample decreased then the water uptake after sample removal is a property characteristic for this exact time and should be calculated as a percent of the sample mass at this exact (not initial) time ((Ww-Wd)/Wd).
- Line 160: proportional to the amount of Ca and P ions
- Line 184: how the Authors define “thermal stability”? In most cases, it is regarded as (a) onset temperature representing first thermal degradation event or (b) temperature representing certain (mostly 5, 10, 50 wt.%) weight lost. As all TG curves are similar, specific values corresponding to the thermal stability should be given. The temperature corresponding to the maximum value of the weight loss rate is not a proper parameter. I could find any values given by the Authors that prove the statement “The thermal stability of cement is improved with the incorporation of CS (Figure 2).” See also lines 283-284.
- Table 2: please present measured values as mean ± standard deviation (SD) (instead of the mean (SD))
- Line 201: biocompatibility is not a reason for changes in WCA value
- Please mark in Fig. 3 by straight line the minimum values specified in ISO 5833-02 (see lines 205-206)
- Line 226: remove “was”
- 6 and 8- the scale of the X-axis should be uniform, numerical. Now this same distance represents 1 week (0-1st and 1st-2nd week),2 weeks (2nd-4th week) and 4 weeks (above 4th week)
- Lines 254-255: please explain, as in my opinion, I could not see the difference in an apatite-like layer – the observed differences are rather a result of an initial surface structure
- Lines 255-256: How the amount of chitosan on the material surface was determined? SEM analysis is not sufficient to do that.
- “The hard glucosamine structure” – what does this statement mean?
- Lines 306-308: verify this sentence as it suggests that amount of oxygen on the surface increases due to the NH2 chitosan hydrophilic groups (see info in bracket)
- Small typing errors should be also revised (e.g. line 232: “20wt.%”)
Author Response
Reviewer 1
Please explain "maximum exotherm temperature" and "setting time" as they cannot be well-known terms for Polymers readers.
R// We appreciate the reviewer's comment, and we added in the results section (lines 192 – 196) an explanation about maximum exotherm temperature and setting time: "The handling parameters that describe the polymerization reaction are maximum temperature (Tmax) and setting time (tset). Tmax corresponds to the maximum temperature reached during the MMA free radical polymerization reaction, which is exothermic, and the tset represents the time taken to reach a temperature midway between ambient and Tmax. From tset, the significant increase of paste's viscosity caused a reduction in the handling of the cement [22]."
The preparation procedure is unclear – first, (line 77) solid constituents (ref. Table 1) were mixed – it suggests that CS and cement powder while the next liquid to solid ratio is given. Additionally, molar/volume/mass ratio?
R// We appreciate the reviewer's comment, and the preparation procedure was rewritten as "ABCs with different CS loadings were prepared. The solid-phase (SP) and liquid phase (LP) were prepared separately by hand mixing, according to the composition shown in Table 1. LP in all formulations was comprised of 95.5 wt.% MMA, 2.0 wt.% of a 50:50 mixture of the comonomers (DEAEA:DEAEM), and 2.5 wt.% DMPT as the activator of the polymerization reaction. LP was added in SP at a solid/liquid ratio = 2 g/ml and manually mixed. The resulting dough was transferred to Teflon molds to obtain specimens to be used in the different tests."
6: in my opinion, if the initial mass of the tested sample decreased then the water uptake after sample removal is a property characteristic for this exact time and should be calculated as a percent of the sample mass at this exact (not initial) time ((Ww-Wd)/Wd).
R// We agree with the Editor's suggestion. Equation 6 was changed, and the values in Figure 6b were recalculated.
Line 160: proportional to the amount of Ca and P ions
R// We appreciate the reviewer's comment, and the sentence "amount of Ca and P ions" was changed by "proportional to the amount of Ca and P ions."
Line 184: how the Authors define "thermal stability"? In most cases, it is regarded as (a) onset temperature representing first thermal degradation event or (b) temperature representing certain (mostly 5, 10, 50 wt.%) weight lost. As all TG curves are similar, specific values corresponding to the thermal stability should be given. The temperature corresponding to the maximum value of the weight loss rate is not a proper parameter. I could find any values given by the Authors that prove the statement "The thermal stability of cement is improved with the incorporation of CS (Figure 2)." See also lines 283-284.
R// We appreciate the reviewer's comment. The text "At temperatures > 400 °C, a significant weight loss of the formulations with lower CS contents is evident" was changed by "The temperature at which a 10 % loss of weight (T10) is achieved is reduced from 304 °C for ABCs without CS, to 288 °C for ABCs loaded with 20 wt.%. Additionally, T50 changed from 377 °C to 380 °C for ABCs without CS and ABC loaded with 20 wt.%, respectively. A 20 CS wt.% introduction decreased the total loss weight percentage to 4°C. "
Table 2: please present measured values as mean ± standard deviation (SD) (instead of the mean (SD))
R// We appreciate the reviewer's comment and the values of Table 2 were changed as mean ± standard deviation
Line 201: biocompatibility is not a reason for changes in WCA value
R// We appreciate the reviewer's comment and the word "biocompatibility" was removed
Please mark in Fig. 3 by straight line the minimum values specified in ISO 5833-02 (see lines 205-206)
R// We appreciate the reviewer's comment, and a straight red line corresponds to the minimum values specified in ISO 5833-02 was added in Figure 3.
Line 226: remove "was"
R// We appreciate the reviewer's comment, and the word "was" was removed.
6 and 8- the scale of the X-axis should be uniform, numerical. Now this same distance represents 1 week (0-1st and 1st-2nd week),2 weeks (2nd-4th week) and 4 weeks (above 4th week)
R// We appreciate the reviewer's comment, Figures 6 and 8 were modified.
Lines 254-255: please explain, as in my opinion, I could not see the difference in an apatite-like layer – the observed differences are rather a result of an initial surface structure
R// We appreciate the reviewer's comment and the expression "except in the appearance of the apatite-like layer deposited on the surface after 20 weeks of soaking, which is more homogeneous and continuous in formulations having CS loading ≤ 10 wt.%" was removed.
Lines 255-256: How the amount of chitosan on the material surface was determined? SEM analysis is not sufficient to do that.
R// We appreciate the reviewer's comment and the expression "The amount of chitosan on the surface increased with an increase in CS loading" was removed.
"The hard glucosamine structure" – what does this statement mean?
R// We appreciate the reviewer's comment, and the word "hard" was changed by "rigid." The statement "the rigid glucosamine structure" means that the cyclic structure of glucosamine reduces the mobility of the MMA chains during the polymerization reaction, thus increasing its viscosity.
Lines 306-308: verify this sentence as it suggests that amount of oxygen on the surface increases due to the NH2 chitosan hydrophilic groups (see info in bracket)
R// We appreciate the reviewer's comment. The expression "because the amount of oxygen present on the surface of the ABC increased due to the hydrophilic groups in the CS (NH2 and OH)" was replaced. The new expression is: "because the greater number of hydrophilic groups (NH2 and OH) provided by the CS."
Small typing errors should be also revised (e.g. line 232: "20wt.%")
R// We appreciate the reviewer's comment and the typing errors were corrected

Reviewer 2 Report
Dear Authors!
Your manuscript overcomes some gaps in chitosan-loaded biomaterials and could be suggested for publication in my opinion.
But there are some misunderstandings in terminology. In my opinion, the terms "bioactivity" and especially "biocompatibility" are not appropriate in the article's content. SBF test has shown the ability of Ca-P deposition (it should be bioactivity but with some limitation and explanation) but biocompatibility used only if you use cell culture or relevant methodologies.
Chatter 2.6 - "In vitro biocompatibility" is not correct. Degradation does not show biocompatibility of any materials - it should be toxic after degradation.
2.6.2 - is not bioactivity
Please correct or answer to this question.
Author Response
Dear Authors!
Your manuscript overcomes some gaps in chitosan-loaded biomaterials and could be suggested for publication in my opinion.
But there are some misunderstandings in terminology. In my opinion, the terms "bioactivity" and especially "biocompatibility" are not appropriate in the article's content. SBF test has shown the ability of Ca-P deposition (it should be bioactivity but with some limitation and explanation) but biocompatibility used only if you use cell culture or relevant methodologies.
R// We agree with the reviewer's suggestion. In the article's content, the word "bioactivity" was modified by "bioactivity in SBF" and the sentence "In vitro biocompatibility" was replaced by "Assessment in a simulated biological fluid (SBF)"
Chatter 2.6 - "In vitro biocompatibility" is not correct. Degradation does not show biocompatibility of any materials - it should be toxic after degradation.
R// We appreciate the reviewer's comment and the sentence "In vitro biocompatibility" was replaced by "Assessment in a simulated biological fluid (SBF)"
2.6.2 - is not bioactivity
R// We appreciate the reviewer's comment, and the word "bioactivity" was modified by the sentence "bioactivity in SBF."
